# Bacterial genome-wide association study of hyper-virulent pneumococcal serotype 1 identifies genetic variation associated with neurotropism

Chrispin Chaguza ⓘ et al.[#]

Hyper-virulent *Streptococcus pneumoniae* serotype 1 strains are endemic in Sub-Saharan Africa and frequently cause lethal meningitis outbreaks. It remains unknown whether genetic variation in serotype 1 strains modulates tropism into cerebrospinal fluid to cause central nervous system (CNS) infections, particularly meningitis. Here, we address this question through a large-scale linear mixed model genome-wide association study of 909 African pneumococcal serotype 1 isolates collected from CNS and non-CNS human samples. By controlling for host age, geography, and strain population structure, we identify genome-wide statistically significant genotype-phenotype associations in surface-exposed choline-binding ($P = 5.00 \times 10^{-08}$) and helicase proteins ($P = 1.32 \times 10^{-06}$) important for invasion, immune evasion and pneumococcal tropism to CNS. The small effect sizes and negligible heritability indicated that causation of CNS infection requires multiple genetic and other factors reflecting a complex and polygenic aetiology. Our findings suggest that certain pathogen genetic variation modulate pneumococcal survival and tropism to CNS tissue, and therefore, virulence for meningitis.

[#]A list of authors and their affiliations appears at the end of the paper.

*S*treptococcus pneumoniae, commonly known as 'the pneumococcus', is a clinically significant opportunistic Gram-positive bacterial pathogen, which causes >320,000 annual deaths in children less than five years old globally, especially in resource poor settings[1,2]. Previous reports have reported that the pneumococcal capsule—which determines the serotype —was a contributing factor to the differential ability of pneumococci to cause invasive pneumococcal diseases such as meningitis—inflammation of the meninges[3–5]. Serotype 1 pneumococci have a high invasive disease-to-carriage odds ratio (>9), highlighting their high invasiveness in contrast with their rare detection rates among asymptomatic individuals[3,6]. While other serotypes such as 7F, 8 and 12F also exhibit relatively high invasive disease-to-carriage odds ratios[6], serotype 1 is ranked as the highest cause of invasive pneumococcal disease in many African countries[7] despite the introduction of the 13-valent pneumococcal conjugate vaccine (PCV), which targets this serotype[8,9]. This feature makes it a unique and prolific public health concern. In Sub-Saharan Africa, serotype 1 disease is endemic[7] and frequently associated with large, rapidly spreading and lethal community meningitis outbreaks[10] with epidemic patterns similar to those seen for *Neisseria meningitidis*[11], both within[12], and outside of the African meningitis belt[11,13,14]. In contrast, small and less severe outbreaks have been detected outside Africa mainly in overcrowded settings such as care homes, schools, and prisons[15–23].

Unlike most other serotypes that rarely cause disease, the high invasive disease-to-carriage odds ratio of serotype 1 strains suggests they resemble a proficient but deliberate invader rather than a typical commensal[3,6,24]. Although serotype 1 strains may be genetically adapted to be more prone at invading their hosts, it remains unknown whether they possess genetic variations other than the capsule biosynthesis genes, which modulates their tissue tropism[25,26]. Such genotypic variants may have a higher ability to migrate and survive in given tissues, such as the cerebrospinal fluid (CSF), where they can cause central nervous system (CNS) pathologies. Previous mutagenesis studies have identified virulence-associated genes and demonstrated their role in colonisation and invasive disease[26–28]. However, well-studied virulence factors, such as pneumolysin, which plays a crucial role in the increased pathogenicity of serotype 1 strains[29], are present in every pneumococcal isolate; therefore, their mere presence/absence patterns are uninformative on the susceptibility and disease severity risk in patients. The analysis of large collections of isolates may reveal with high resolution the existence of previously unknown genetic variations not only in terms of presence/absence of genes but also single-nucleotide polymorphisms (SNPs) and insertions/deletions, which may contribute to the pathogen virulence to causing certain invasive diseases.

Genome-wide association studies (GWAS) are increasingly used to investigate the statistical link between genotypic variation and bacterial phenotypes[30,31]. Previous studies identified genetic variation linked with disease susceptibility[32–38], nutrient synthesis[39], carriage duration[40], disease progression[32], host adaptation[41], virulence[42] and antimicrobial resistance[43–47]. However, GWAS analyses of bacterial isolates sampled from different tissues have yielded inconsistent findings on the contribution of genetics on tissue tropism and disease susceptibility[32–37]. This is exemplified by studies comparing carriage and disease isolates, which have identified variants associated with invasiveness[34,38,48] while studies comparing invasive isolates from different tissues, such as blood and CSF, have yielded no differences[32,33,37]. These inconsistencies reflect differences in the analytical methods, data set sizes, geographical settings and control for confounders, such as capsular diversity, geographical origin and strain population structure. The latter is especially problematic in bacterial species with highly structured populations[44], but may be less severe in highly recombinogenic species such as *S. pneumoniae* in which the genetic pool is frequently shuffled[49].

We have amassed a large collection of 909 invasive pneumococcal serotype 1 isolates from Sub-Saharan Africa predominantly belonging to the clonal complex (CC) 217, a dominant and endemic hyper-virulent lineage on the continent[50]. The isolates were collected between 1996 and 2016, and were sequenced within the framework of multi-national genomic surveillance consortium studies; the Pneumococcal African Genomics (PAGe)[51] and the Global Pneumococcal Sequencing (GPS) project[52]. In this study, we conducted GWAS analysis to compare serotype 1 isolates sampled from CNS and non-CNS human specimens to determine whether presence of genetic variation is disproportionately enriched in the CNS isolates, which may contribute to CNS tissue neurotropism—the ability of the pneumococci to translocate across the blood–brain barrier to cause meningitis. We used three types of genetic variation for the GWAS namely, presence/absence patterns of accessory genes, unique DNA substrings of variable length (or unitigs)[53], and single-nucleotide polymorphisms (SNPs). We employed robust linear mixed model approaches to control for important covariates, including, host age, geographical origin, capsular diversity and fine scale strain population structure. We also assessed the phylogenetic and geographical distribution, heritability and biological relevance of the identified variants. Our study uncovers the existence of genetic variants, which modulates the propensity of pneumococcal strains to translocate to, survive or resist immune clearance in the CSF, thus highlighting potential targets for therapeutic and prophylactic interventions to prevent and control pneumococcal diseases.

## Results

**Characteristics of the serotype 1 isolates**. We compiled data for 909 *S. pneumoniae* serotype 1 isolates originating from Sub-Saharan Africa and carried out a GWAS analysis to determine genetic differences between CNS and non-CNS isolates (Supplementary Data 1). Of these isolates, 297 were isolated from CSF in patients while 612 non-CNS isolates were isolated from the blood, lung aspirate, joint fluid, and pleural or peritoneal fluid (Fig. 1a). All the CSF specimens were cultured from patients with meningitis or other CNS infections. The isolates were collected via hospital-based bacterial surveillance between 1996 and 2016. Age is an essential confounding factor associated with co-morbidities, immaturity and senescence of the immune system, which contributes to patient susceptibility to CNS invasion. We recorded age as a continuous variable, but the number of the patients by age groups were as follows; <2 years old ($n = 158$), 2–4 years ($n = 209$), 5–15 years ($n = 171$), >15 years ($n = 255$) and age unknown ($n = 116$) (Fig. 1b). The patients were from eleven countries namely; Ghana ($n = 2$), Ivory Coast ($n = 1$), Malawi ($n = 197$), Niger ($n = 43$), Nigeria ($n = 6$), Senegal ($n = 12$), South Africa ($n = 357$), The Gambia ($n = 181$), Togo ($n = 44$), Benin ($n = 2$) and Mozambique ($n = 64$) (Fig. 1c). We identified 34 sequence types (ST) inferred from whole-genome sequences using the multilocus sequence typing (MLST)[54], which belonged to the hyper-virulent African clonal complex (CC) 217 lineage[50,51]. The most common clones based on STs were ST217 ($n = 524$), ST3081 ($n = 113$), ST612 ($n = 69$), ST618 ($n = 50$), ST303 ($n = 44$) and ST11745 ($n = 11$) (Fig. 2). All the isolates corresponded to the GPSC2 lineage based on the Global Pneumococcal Sequence Cluster (GPSC) nomenclature as the definition for international pneumococcal lineages[52].

**Phylogenomic and geographical diversity of the isolates**. We performed population structure analysis to understand the

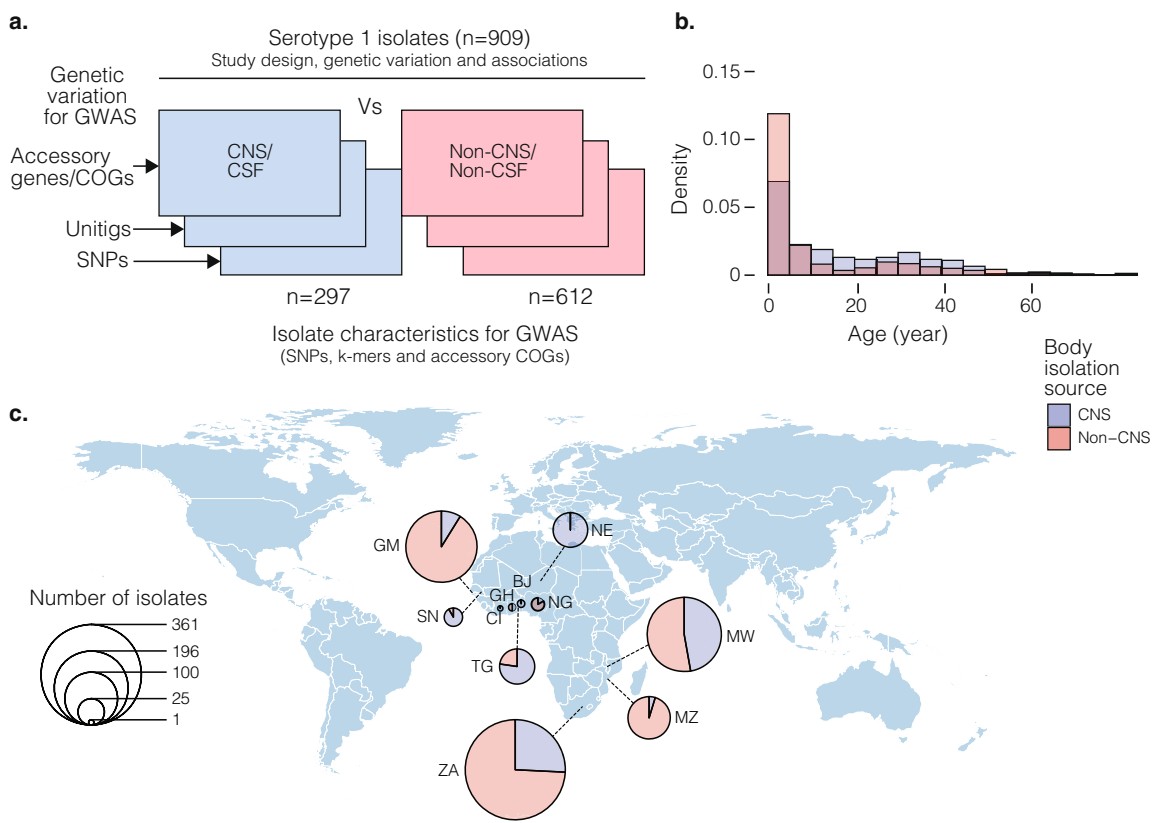

**Fig. 1 Characteristics of the African *S. pneumoniae* serotype 1 isolates. a** Study design of the pathogen genome-wide association study (GWAS) showing the number of the central nervous system (CNS) and non-CNS isolates and three types of genetic variation namely single-nucleotide polymorphisms (SNPs), unitigs and accessory clusters of orthologous genes (COGs) used for the analysis. **b** Histogram showing age distribution of patients whose CNS and non-CNS isolates were sampled. The two histograms are coloured by isolation source whereby darker shades indicate an overlap. **c** Country of origin of the isolates, their frequency and proportion of the CNS and non-CNS isolates at each location are shown as pie charts. The size of the pie charts is proportional to the number of isolates from each country as shown by the scale represented by the concentric circles at the bottom left of the diagram. The country names are designated by their international two letter codes as follows: South Africa (ZA), Malawi (MW), The Gambia (GM), Ghana (GH), Niger (NE), Nigeria (NG), Togo (TG), Benin (BJ), Côte d'Ivoire or Ivory Coast (CI) and Senegal (SN). All the metadata associated with the isolates in the phylogeny are provided in the appendix (Supplementary Table 1) while data shown in the figure is shown in Supplementary Data 2.

genetic diversity of the isolates (Fig. 2 and Supplementary Fig. 2). Bayesian clustering yielded seven monophyletic clades 1–7. We then used the Chi-squared ($\chi^2$) test to assess the phylogeographic structuring of the isolates and found a strong association ($P < 2.2 \times 10^{-16}$) between the clades and country of origin consistent with the phylogenetic patterns observed in previous studies of African serotype 1 isolates based on STs. Based on the Kruskal–Wallis test, there were significant heterogeneities between the clades in genetic diversity ($P < 2.2 \times 10^{-16}$), geographical variability ($P < 2.2 \times 10^{-16}$), and Simpson diversity in terms of country of origin ($P < 2.2 \times 10^{-16}$) and MLST ($P < 2.2 \times 10^{-16}$) (Fig. 3a–d). Clades 3, 6 and 7 are geographically diverse with some geographically distant isolates separated by relatively few SNPs (10 to 50), suggesting recent geographic transmission (Fig. 3e and Supplementary Fig. 3). This demonstrates the need for adequate control of geographical variation and strain population structure in the GWAS analysis.

**GWAS reveals unitigs associated with CNS isolates.** We applied linear mixed model approaches (implemented in FaST-LMM[55] and GEMMA[56]) to test for genetic associations with CNS and non-CNS disease (Fig. 3) using three types of genomic variation namely, SNPs, accessory gene content and unitigs[53]. Unitigs are increasingly popular for bacterial GWAS analysis because they

capture SNP, and insertion and deletions (indels) simultaneously in both coding and non-coding regions[53,57]. We identified 123,401 unitigs sequences from the assemblies of the 909 isolates. We generated a matrix showing presence/absence patterns of each unitig in the isolates and then we filtered out low frequency unitigs with minor allele frequency (MAF) < 1% resulting in a reduced matrix with 20,673 unitigs for the GWAS analysis.

With these unitigs we used a univariate linear mixed model to test for associations between the presence and absence of a unitig and CNS infection whilst controlling for population structure in terms of the kinship matrix, and host age and geographical origin as covariates. The resulting QQ-plots confirmed adequate control of the population structure (Supplementary Fig. 4). Capsular type diversity is effectively removed as a potential confounder as the analysis is focused on a single pneumococcal serotype[58]. We identified two genome-wide significant unitigs using FaST-LMM whose presence/absence were associated with CNS isolates (Fig. 4a). The first unitig; ID 8805 (odds ratio = 0.70, $P = 5.00 \times 10^{-08}$), was associated with pneumococcal surface protein C (*pspC*) gene, also known as choline-binding protein A (*cbpA*) or *spsA* (Figs. 4a and 5a, Table 1 and Supplementary Table 2 and Fig. 5). This unitig mapped to the proline-rich surface-exposed region

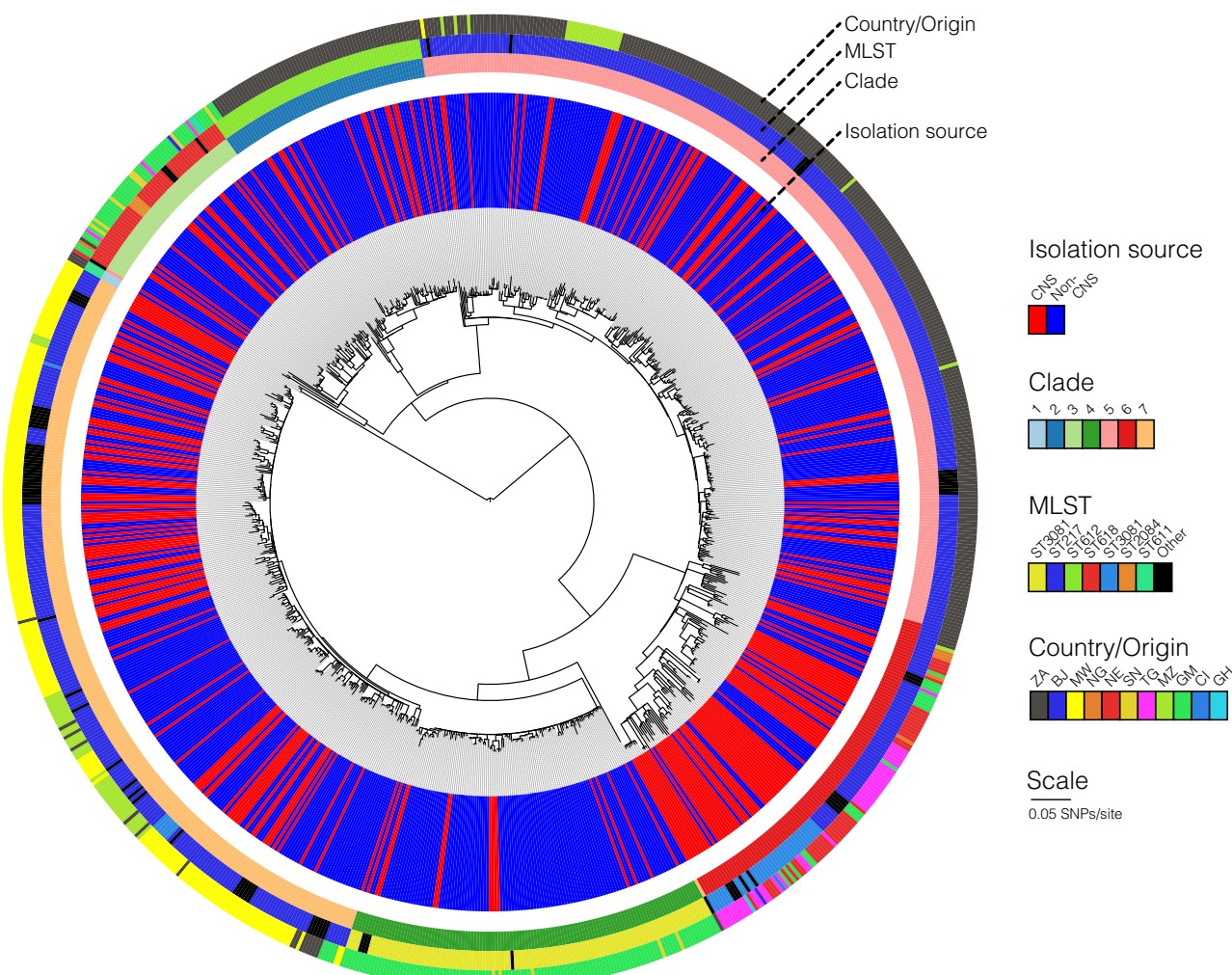

**Fig. 2 Whole-genome phylogenetic tree showing genetic similarity of the 909 African *S. pneumoniae* serotype 1 isolates.** A mid-point rooted whole-genome phylogenetic tree depicting the genetic relatedness of the isolates after filtering out genomic regions with recombination. The coloured strips at the tips of the tree indicates isolate metadata: ST, isolation source and country. The country names are designated by their international two letter codes as follows: South Africa (ZA), Malawi (MW), The Gambia (GM), Ghana (GH), Niger (NE), Nigeria (NG), Togo (TG), Benin (BJ), Côte d'Ivoire or Ivory Coast (CI) and Senegal (SN). All the metadata associated with the isolates in the phylogeny are provided in the appendix (Supplementary Table 1) while data shown in the figure is shown in Supplementary Data 2.

of *pspC*, which elicits antibody-mediated protection[59], which could explain why the absence of this variant was less common in CNS isolates. The second significant unitig ID 47853 (odds ratio = 0.71, $P = 1.32 \times 10^{-06}$) was associated with the putative DnaQ family exonuclease or DinG family helicase gene, whose role in pneumococcal pathogenesis is unknown. GWAS analysis using GEMMA also detected significant association with the unitig ID 8805 (odds ratio = 0.70, $P = 4.76 \times 10^{-08}$) but only FaST-LMM detected the variant in DnaQ family exonuclease as significant while GEMMA reported it as a suggestive hit (Fig. 4 and Table 1). We also detected a further 15 unitigs in genes and intergenic with *P*-values above the suggestive threshold. Some of the genome-wide significant and suggestive unitigs showed significant correlation patterns (Supplementary Fig. 6).

**GWAS analysis of the presence/absence of accessory genes.** Sequence clustering of coding sequences from the entire data set revealed a pan-genome comprising 5759 clusters of orthologous genes (COG). To correct potential errors caused by skipped gene model prediction and annotation, representative COG sequences for each cluster were compared to the draft assemblies to refine

presence/absence patterns of the genes. After filtering out the COGs with MAF < 1%, 1068 COGs were selected and subjected to the GWAS analysis using the same linear mixed model approaches as the ones used for the unitig-based GWAS. There were no genome-wide significant COGs, however, the *P*-value estimated by GEMMA (odds ratio: 1.10, $P = 7.34 \times 10^{-04}$) for a single COG (ID: 445) was above the threshold for suggestive hits (Fig. 4b). This suggestive COG coded for a type 1 restriction modification system (RMS) subunit S protein, which plays several roles, including regulation of capsule production[60]. Assessment of the observed and expected *P*-values suggested the difference was not due to population structure (Supplementary Fig. 4). Genomic annotation of this COG 445 showed that it encodes a 60 amino acid type 1 RMS protein located adjacent to the *hsdM* protein in the chromosome, which collectively constitute a reversible phase variable locus in the pneumococcus[61–63]. Additional genomic analysis revealed that harbouring a truncated non-functional version of this gene appeared to be common among non-CNS rather than CNS isolates in some lineages although it did not reach genome-wide but was above the suggestive threshold (Supplementary Fig. 7).

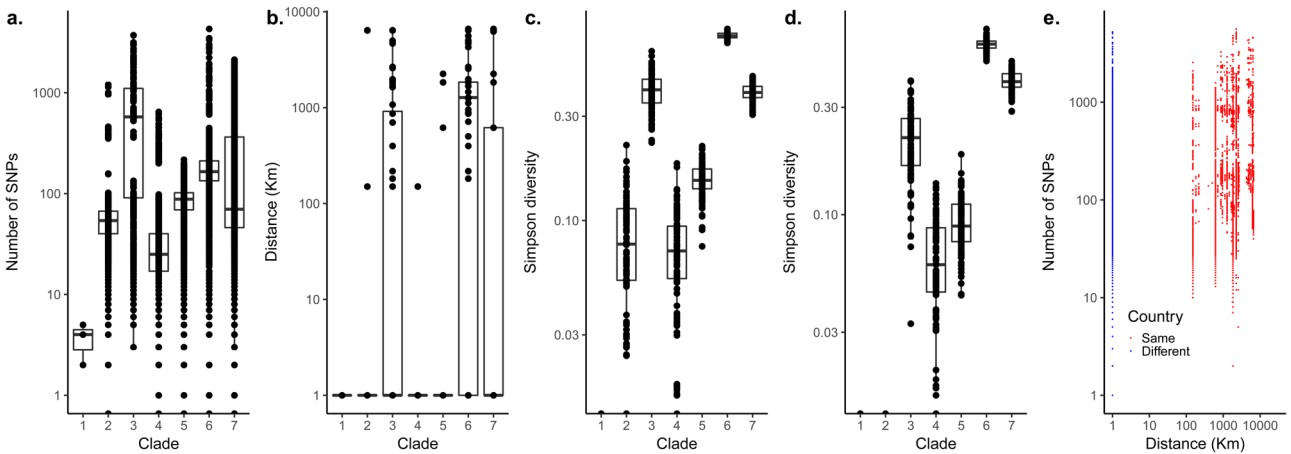

**Fig. 3 Genetic diversity of the African *S. pneumoniae* serotype 1 isolates.** Boxplots overlaid with dot plots showing the distribution of **a** the average number of SNPs between isolates in each clade, **b** geographical distance between pair of isolates in each clade, **c** the Simpson diversity index values for the composition of the isolates in each clade by country of origin and **d** Simpson diversity index values for the composition of the isolates in each clade by sequence type (ST). **e** Scatter plot showing the relationship between the number of SNPs and geographical distance (in kilometres [Km]) between pair of isolates. Both axes are shown in logarithmic scale (base 10) for clarity. The points coloured in blue in panel **e** represent isolates from the same country while those coloured in red represent isolates from different countries. The density of the points on each axis of the graph are represented by the dashed lines at the top and far right of the scatter plot. Further breakdown of the plot **e** is provided in the appendix (Supplementary Table 1) while data shown in the figure is shown in Supplementary Data 2.

**Fig. 4 SNPs, unitigs and accessory genes associated with CNS infection. a** show statistical significance (–-log₁₀P) for the unitigs (unordered) from the pathogen genome-wide association study (GWAS) analysis using FaST-LMM and GEMMA. **b** show the statistical significance of the accessory clusters of orthologous genes (COGs) using the same methods. **c** show the chromosomal locations and statistical significance of the single-nucleotide polymorphisms (SNPs). The green and red lines designate the genome-wide significant and suggestive *P*-value thresholds as discussed in the 'methods' section while the variants highlighted in yellow were identified by both FaST-LMM and GEMMA methods. All the variants had minor allele frequency (MAF) < 1% and missingness >5%. The data shown in the figure is shown in Supplementary Data 2.

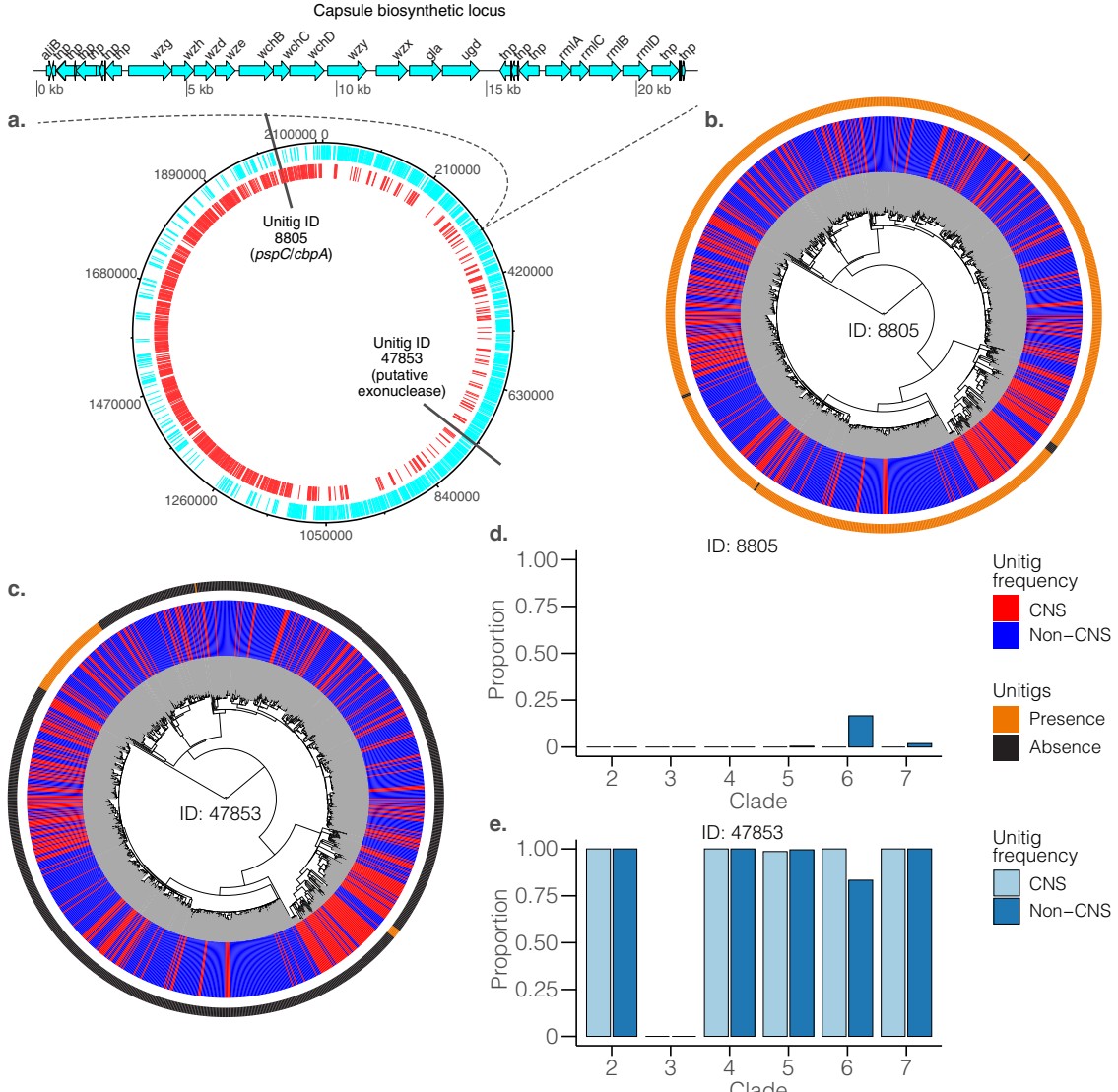

**Fig. 5 Phylogenetic distribution of the genome-wide significant and suggestive unitigs. a** Schematic representation of the serotype 1 genome showing location of the capsule biosynthetic locus and the identified genome-wide significant unitigs in the P1031 serotype 1 reference genome (GenBank accession: CP000920) with coding sequences reannotated with common gene names. **b, c** Circular phylogenetic trees showing the distribution of the two genome-wide significant unitigs ID 8805 and 47853. **d, e** Frequency of the unitigs in CNS and non-CNS isolates in each phylogenetic clade defined in Fig. 2. The full list of the genome-wide significant unitigs represented by the patterns shown in this figure is provided in the appendix (Supplementary Table 1) while data shown in the figure is shown in Supplementary Data 2.

**Suggestive associations detected by single base nucleotide changes**. We identified 45,083 SNPs from whole-genome alignment of the serotype 1 isolates created via consensus sequences after mapping the sequence reads of each genome against a high-quality African serotype 1 reference genome. After filtering out SNPs with MAF < 1% and missingness >5%, we were left with 2393 SNPs for the association analysis using the same tools and controlling for the same covariates as for the COG and unitig-based analysis. There were no SNPs with $P$-values below the genome-wide statistical significance threshold (Fig. 4c and Table 2). However, both GEMMA and FaST-LMM identified two suggestive SNPs but none of them were identical. All the suggestive SNPs were in the intergenic regions (Table 2).

**Effect sizes, heritability and distribution of the significant variants**. The effect sizes for the effect/minor alleles the genome-wide significant unitigs identified by FaST-LMM and GEMMA

were −0.075 (odds ratio: 0.699) and −0.178 (odds ratio: 0.708), respectively. This implies that the absence of these unitigs was associated with decreased propensity of the bacteria to cause CNS infection (Table 1). However, the overall heritability was almost zero, which implied that despite the presence of individual variants associated with causation of CNS infections, the phenotypic variation explained by the overall pathogen genetics was inadequate to explain the phenotype. Since the phylogeny of the isolates showed a strong phylogeographic structure (Fig. 2), we then assessed the distribution of the two genome-wide significant unitigs: IDs 8805 and 47853 by clade, which were associated with *pspC* and the putative DnaQ exonuclease protein respectively. We observed 100% (79/79) frequency of unitig ID 8805 in clade 6 but there was lower frequency of 83.3% (30/36) in non-CNS isolates. However, the frequency was nearly identical between CNS and non-CNS isolates compared to the other clades (Fig. 5). The overall frequency of unitig ID 47853 across the entire data set was lower than unitig ID 8805 but was detected in non-CNS isolates

**Table 1 Summary of the genome-wide significant and suggestive unitigs associated with CNS infection identified by the GWAS analysis.**

| Unitig | Gene | Genome accession | Locus tag | Risk allele | MAF | Odds ratio | P-value | Gene description |
|---|---|---|---|---|---|---|---|---|
| 8805[a] | pspC | CP000936.1 | SPH_2388 | Absence | 0.01 | 0.70 | $5.0 \times 10^{-08}$ | Surface protein PspC |
| 47853[a] | | NC_014498.1 | SP670_1521 | Absence | 0.07 | 0.71 | $1.3 \times 10^{-06}$ | DnaQ exonuclease/DinG helicase family |
| 41314 | glmM | NC_011072.1 | SPG_1486 | Absence | 0.03 | 1.18 | $1.2 \times 10^{-05}$ | Phosphoglucosamine mutase |
| 72152 | | AP018043.1 | Intergenic region | Absence | 0.02 | 1.24 | $3.5 \times 10^{-05}$ | Intergenic region |
| 80564 | cca | CP000920.1 | SPP_1579 | Presence | 0.03 | 1.19 | $3.0 \times 10^{-05}$ | tRNA nucleotidyltransferase |
| 81567 | | AP019192.2 | ASP0581_08080 | Presence | 0.01 | 0.76 | $3.9 \times 10^{-05}$ | cysteine desulfurase |
| 90414 | glmM | NC_017592.1 | SPNOXC_13690 | Presence | 0.03 | 1.18 | $1.2 \times 10^{-05}$ | Putative phosphoglucosamine mutase |
| 102497 | | AKBW01000001.1 | Intergenic region | Absence | 0.07 | 1.14 | $9.1 \times 10^{-06}$ | Intergenic region |
| 102498 | | AP017971.1 | KK0981_35330 | Presence | 0.07 | 1.14 | $1.1 \times 10^{-05}$ | Cytoplasmic protein |
| 106507 | | AP017971.1 | KK0981_35330 | Presence | 0.07 | 1.13 | $2.4 \times 10^{-05}$ | Cytoplasmic protein |
| 108518 | | AP017971.1 | KK0981_35330 | Presence | 0.07 | 1.13 | $2.9 \times 10^{-05}$ | Cytoplasmic protein |
| 110000 | | NC_014498.1 | SP670_1747 | Presence | 0.01 | 0.75 | $1.6 \times 10^{-05}$ | Hypothetical protein |
| 47853 | | NC_014498.1 | SP670_1521 | Presence | 0.07 | 0.71 | $1.3 \times 10^{-06}$ | DnaQ exonuclease/DinG helicase family |
| 47851 | | AP018043.1 | KK0381_02650 | Absence | 0.07 | 0.70 | $3.0 \times 10^{-06}$ | Bifunctional ATP-dependent DNA helicase |
| 108605 | | NC_014498.1 | SP670_1521 | Presence | 0.07 | 0.70 | $3.0 \times 10^{-06}$ | DnaQ exonuclease/DinG helicase family |
| 45790 | | AP019192.2 | ASP0581_14110 | Presence | 0.47 | 0.78 | $9.4 \times 10^{-06}$ | N-acetyltransferase |
| 70431 | | AP019192.2 | ASP0581_14110 | Presence | 0.47 | 0.76 | $1.4 \times 10^{-05}$ | N-acetyltransferase |
| 102497 | | AKBW01000001.1 | NA | Absence | 0.07 | 1.14 | $2.1 \times 10^{-05}$ | Intergenic region |

The risk genotype refers to the minor allele used as the effect/non-reference allele.
The likelihood ratio P-values shown were estimated by the method, which detected the variant as genome-wide significant or suggestive. When both GEMMA and FaST-LMM identified variants as statistically significant, P-values from GEMMA were used.
MAF minor allele frequency.
[a]Genome-wide significant unitigs.

**Table 2 Summary of the suggestive SNPs identified by the GWAS analysis.**

| SNP | Genomic region | Allele (risk/safe) | MAF | Odds ratio | P-value | Gene | Gene description |
|---|---|---|---|---|---|---|---|
| rs266945 | Genic | A/G | 0.097 | 1.29 | $4.67 \times 10^{-05}$ | glmS | Glutamine-fructose-6-phosphate transaminase (isomerising) |
| rs721084 | Genic | A/G | 0.042 | 1.19 | $1.05 \times 10^{-04}$ | | Membrane-fusion protein |
| rs1466695[a] | Genic | A/G | 0.015 | 1.22 | $3.41 \times 10^{-04}$ | glmM | Phosphoglucosamine mutase |
| rs1474820[a] | Genic | G/A | 0.036 | 1.14 | $4.08 \times 10^{-04}$ | clpX | ATP-dependent Clp protease, ATP-binding subunit ClpX |

The SNPs were identified and annotated using the S. pneumoniae serotype 1 reference genome strain P1031 (GenBank accession: CP000920).
The risk genotype refers to the minor allele used as the effect/risk/non-reference allele while the safe allele refers to the reference allele/genotype in the GWAS analysis.
MAF minor allele frequency.
[a]SNP identified by FaST-LMM.

only. Unitig ID 47853 was found in clade 5, 6 and 7 with a prevalence of 0.56% (1/218), 16.7% (6/36) and 1.89% (3/159), respectively (Supplementary Fig. 8).

## Discussion

Despite the rollout of 13-valent PCV in Sub-Saharan Africa[8,9], serotype 1 pneumococci remain a significant cause of life-threatening invasive pneumococcal diseases including meningitis[7,12,64]. By leveraging a large collection of hyper-virulent pneumococcal serotype 1 clinical isolates, we have discerned the contribution of pathogen genetics to the propensity of the strains to cause CNS infections. Through a linear mixed model GWAS analysis, we have identified genome-wide statistically significant genetic variation differentially abundant in CNS and non-CNS isolates in genes, which encodes a surface-exposed protein (pspC or cbpA) and a hypothetical DnaQ/DinG exonuclease/helicase family gene with unknown function. The PspC protein is a multifunctional choline-binding protein, which binds to human factor H[65], complement inhibitor C4b[66] and inhibits complement C3 deposition[67], which promotes immune evasion and virulence[67,68]. Crucially, experimental studies have shown that interactions between PspC and the C-terminus of the laminin-integrin receptor, initiates contact with the vascular endothelium of the blood–brain barrier, which improves pneumococcal tropism to the CNS[69], a mechanism also utilised by other respiratory bacterial pathogens[69], neurotropic viruses[70–73] and prions[74]. PspC also plays a role in pneumococcal translocation across epithelial surfaces, which may play a crucial role during infection[75]. Furthermore, PspC binds the polymeric immunoglobulin receptor (pIgR) mediating trans-cellular transport[76] and it is up-regulated upon contact with epithelial cells[77] while its genetic diversity has been linked with susceptibility to invasive diseases[34], immune evasion[78] and variable invasiveness[79]. Therefore, the

identified allelic variation in the proline-rich repeat region of *pspC* from our GWAS analysis results in differential propensity of the strains to translocate across the blood–brain barrier to the CNS to cause meningitis by modulating pneumococcal interaction with the laminin-integrin receptor, thus, highlighting *pspC* as a novel target for clinical interventions. The absence of *pspC* in *Streptococcus mitis* and other commensal streptococci[80] and high sequence conservation of the genomic region harbouring the unitig in *pspC* (Supplementary Data 3) further enhances its potential use in a vaccine[81,82].

Since *pspC* is a known hotspot for homologous recombination[49], it's likely that the identified variants were generated via genetic exchange but unstable tandem repeat regions in non-alpha helical regions may equally contribute to its allelic diversity as a surface-exposed protein encoding gene[59]. While the present findings support PspC as a potential anti-virulence vaccine candidate, due to the inherent genetic variability of the gene[78], vaccines based on this protein would require targeting invariant regions of the gene to achieve better protection[83]. Since the gene encoding for the DnaQ exonuclease/DinG helicase family protein has not yet been characterised by previous studies, its specific role on pneumococcal disease pathogenesis remains unknown. Therefore, this study provides a platform for further studies to gain insights into the biological function of the identified proteins and how their allelic variation predisposes pneumococcal strains to cause meningitis at an increased propensity.

A recent GWAS study based on the Active Bacterial Core surveillance (ABCs) data set of pneumococcal serotypes detected in the USA has associated the *pbp1b*641C missense mutation with pneumococcal meningitis[35]. However, no variants detected in our study contained the *pbp1b*641C mutation, which would imply that it is biologically favourable in the CC217 background. Geographically, CC217 is primarily restricted to Sub-Saharan Africa, therefore, the rarity of this clone in the USA could also explain why none of the variants detected in this study were detected in the ABCs collection as these may be specific to the CC217 background[50,84]. Indeed, the effect of phylogeography and lineage were also evident in our analysis reflecting the impact of local spatial selection pressures on the susceptibility to CNS infections[51]. Altogether, our study suggests that geographical differences such as local selection pressures, e.g., driven by antibiotic use, may result in local emergence and circulation of pneumococcal strains containing genetic predispositions for CNS infections. As such, additional studies are required to assess the presence of risk genetic variants in different geographical regions globally.

The small effect sizes in our study for the genome-wide significant variants; odds ratio: 0.6 to 1.8, are consistent with estimates elsewhere in bacteria[34,40] and complex human diseases and traits[85]. This suggests that the variants have been shaped by neutral selection unlike the variants associated with antimicrobial resistance, which are typically under strong selection pressure[43–45,86]. However, larger effect sizes (odds ratio >3) for disease susceptibility have also been reported in bacterial GWAS studies in variants associated with susceptibility to Staphylococcal pyomyositis[36] and clinical manifestation of pneumococcal infections[87], suggesting that although uncommon, disease risk loci may also be under selection. The *pbp1b*641C missense mutation associated with pneumococcal meningitis with high effect size in the ABCs data set in the US could be because the variant promotes tolerance to penicillin, and is therefore likely to be subject to selection[35]. Therefore, since invasive disease is an evolutionary dead-end, the genetic variants associated with disease phenotype, such as those found in this study, are unlikely to be subject to positive selection barring additional unknown functions on carriage and transmission.

In contrast to previous bacterial GWAS studies assessing niche-specific differences between isolates across the multiple serotypes and lineages[33,37], our study focused on a well-sampled clinically relevant serotype with diverse geographical representation from both West and Southern Africa, and from patients with a wide range of syndromes. This removed the effect of capsular diversity in the GWAS analysis, which has affected similar bacterial GWAS studies[37] (Table 2). With the linear mixed model GWAS, important confounders such as strain population structure, unbalanced sampling of CNS and non-CNS isolates by country, host age and geographical origin of the isolates were controlled for, which combined with analysis of multiple types of genetic variation; SNPs, accessory genes and unitigs, makes our analysis robust and comprehensive. Although some isolates were sampled after the introduction of the 13-valent PCV, we are of the view that vaccination is unlikely to have biased our GWAS analysis as the target for the PCVs is the polysaccharide capsule (identical in all isolates) and not protein variants outside the capsule biosynthetic locus such as those identified by our analysis. However, since the CNS isolates share a common intermediate route of infection with non-CNS isolates especially in the lung and blood, some of the non-CNS isolates may have been captured early en route to the CSF thereby resulting in misclassification. Although the extent of this is not known, we speculate its effect to be minor partly because there is typically delay in seeking and initiating treatment among patients with serious bacterial infections in our study setting in contrast to high-income settings[88]. This implies that the majority of the isolates definitely en route to the CNS may have been sampled post-translocation into the CSF rather than in intermediate tissues such as blood. Similar sampling issues have been encountered in studies elsewhere comparing bacterial isolates from different tissues[33,48,89]. While the power to detect genetic differences between CNS and non-CNS isolates may have been partially obscured because of these issues, the identification of genome-wide significant hits indicates that the effort to detect such variants is not futile. The analysis of additional sample collections might allow uncovering of additional hidden variants.

The GWAS approach used in this study negates the need for specifying hypotheses regarding candidate genes prior to the analysis. While our study has identified promising hits, which individually are associated with propensity for causing CNS infection, however, the negligible heritability suggests that the overall pathogen genetics may be insufficient to explain the phenotypic variability, which reflect the complexity and polygenic aetiology of CNS infection. Causation of CNS infections requires an interplay of host-pathogen interactions, and the effect of many cryptic genetic variants with small effect sizes, low penetrance and expressivity, and possibly weak epistatic effects. Our study has identified targets for follow-up in vitro and in vivo phenotypic studies to validate their contribution to the pathogenesis of pneumococcal CNS infections. Altogether with the findings on the elicitation of protective antibody by the identified proline-rich variants in *pspC*[59] and modulation of pneumococcal tropism across the blood–brain barrier to CNS via the PspC-laminin-integrin receptor binding[69], our findings provide further evidence to inform the design of protein-based vaccines to prevent pneumococcal CNS infections such as protein-based vaccines. Such vaccines can be used as standalone or adjuvants in conjunction with 13-valent PCV, whose effectiveness against pneumococcal serotype 1 seems either inadequate or delayed[8,9], hence these would be particularly effective when deployed in reactive vaccination campaigns during serotype 1 meningitis outbreaks in Sub-Saharan Africa.

## Materials and methods

**Sample characteristics and preparation**. Invasive *S. pneumoniae* samples ($n = 909$) were collected from the central nervous system (CNS), i.e., cerebrospinal fluid (CSF), and non-CNS tissues from hospitalised patients of any age through hospital-based surveillance conducted by the collaborating institutions, including the main partner African centres; Malawi-Liverpool-Wellcome Trust Clinical Research Programme (MLW) in Malawi, the National Institute for Communicable Diseases (NICD) in South Africa, Medical Research Council (MRC) Unit at the London School of Hygiene and Tropical Medicine (LSHTM) in The Gambia, Centro de Investigação em Saúde da Manhiça in Mozambique and Centre de Recherche Médicale et Sanitaire in Niger (Supplementary Data 1). Samples from West African countries were collected through the WHO collaborating Centre for New Vaccines Surveillance at MRC The Gambia at the London School of Hygiene and Tropical Medicine, which supports Invasive Bacterial Vaccine-Preventable disease surveillance (IBVPD) in the region[90]. Isolates associated with known meningitis outbreaks, particularly in West Africa, and those with either unknown source of isolation or non-specific information on sampling locations, e.g., those recorded as sampled from either blood and CSF or broadly as invasive, were excluded to avoid ambiguity between cases and controls in the GWAS analysis. We cultured the isolates and extracted genomic DNA using approaches described elsewhere[91,92].

**DNA sequencing, assembly and isolate typing**. The extracted DNA was shipped to the Wellcome Sanger Institute, where genomic DNA libraries were prepared for WGS using Genome Analyser II and HiSeq 4000 Sequencing Systems (Illumina, CA, USA). The following quality control (QC) criteria was used: samples were only included when the overall sequencing depth was >75x, percent coverage of reads of >80% across the pneumococcal genome (GenBank accession number: FM211187), an assembly length was between 1.9 and 2.3 Mb, and the number of assembled contigs was <200. Samples with 15% of heterozygous SNP sites over the total SNP sites and >25% maximum minor allele frequency (MAF) were suggestive of a mixture of two or more *S. pneumoniae* isolates in a single-DNA sample and were therefore also excluded. The median length of the sequence reads was 100 and the average quality score was 34.8. The reads were de novo assembled into contiguous sequences using an automated pipeline[93]. The detailed genome assembly metrics are shown in Supplementary Fig. 1.

Serotyping of the pneumococcal isolates was done using k-mer based in silico serotyping based on genomic data using SeroBA v1.0.0[94]. Pneumococcal clones or sequence types (ST) were inferred using the pneumococcal multilocus sequence typing (MLST) scheme[54] implemented in MLSTcheck v2.0.1510612[95], while the international pneumococcal lineage nomenclature defined by the global pneumococcal lineage clusters (GPSC)[52] were detected using PopPUNK v1.1.7[96]. The consensus whole-genome sequence alignment was generated by mapping reads to the complete reference genome of a serotype 1 pneumococcal ST303 strain P1031 (GenBank accession: CP000920) using SMALT v0.7.4 [www.sourceforge.net/projects/smalt/] (minimum insert size: 50, maximum insert size: 1000, minimum quality: 30, minimum depth of coverage: 4, minimum matching reads per strand: 2 and minimum base quality: 50) (Supplementary Fig. 1). The insertions and deletions were realigned using GATK v4.0.3.0[97].

**Phylogenetic analysis and SNP patterns**. The generated whole-genome alignment was iteratively screened for recombination events for removal using Gubbins v1.4.10[98], prior to construction of a maximum likelihood phylogeny with RAxML v7.0.4[99] using GTR + $\Gamma$ (gamma) model[100] and 100 bootstrap replicates[101]. Visualisation and annotation of the phylogeny was done using the iTOL v2.0[102]. The multiple sequence alignment and variant call format (VCF) for the sites with single-nucleotide polymorphisms (SNP) in the whole-genome alignment were generated using Snp-Sites v2.3.2[103]. The VCF files were used to generate PLINK-formatted pedigree files for the genome-wide association study (GWAS) analysis using VCFtools v0.1.16[104]. The pedigree files were merged with phenotype (CNS and non-CNS isolation status) and variants with minor allele frequency (MAF) < 1% and missingness >5% were filtered out using PLINK v1.90b4[105]. Phylogenetic clades were inferred from the 45,083 bp SNP alignment of the isolates using fas-tBAPs v1.0.0[106] using the 'baps' prior optimisation option. Before running fast-BAPS, we filtered out non-informative and private mutations from the SNP alignment using *extract_PI_SNPs.py* script (https://gist.github.com/jasonsahl/9306cd014b63cae12154). Genetic diversity of the isolates in each clade were assessed using the Vegan v2.5.4 package[107] while the geographical distance was calculated using the 'distHaversine' function in 'geosphere' v1.5.7 package in R v3.5.3 (R Foundation Core Team, 2019). Where >2 alleles were detected at each genomic position, we generated bi-allelic variants at these positions by selecting only the two most common nucleotides. We used two of the most common nucleotides detected at each position variants at each chromosomal position for the analysis. Genomic location of genes were determined using nucleotide BLAST v2.2.30 + [108] and visually assessed using ACT v9.0.5[109] and DNAPlotter v1.0[110] while multiple sequence alignments were generated using MUSCLE v3.8.31[111].

**Pan-genome gene presence/absence patterns**. The draft genomic assemblies were annotated using Prokka v1.11[112]. These annotated assemblies were processed by Roary v3.6.1 pan-genome pipeline[113] to identify the clusters of orthologous genes (COGs). We selected the reference gene for each COG and compared this to each draft assembly using BLAST (percent identity: 85%, query coverage: 85%) in order to generate a COG presence/absence matrix. We then checked for unique COG presence-patterns and then created a reduced COG matrix by filtering out COGs with MAF and missingness <1% and >5% respectively and combined it with the phenotypic data (disease status) into the PLINK-formatted pedigree files for GWAS. We validated the pedigree files MAF and missingness filtering using PLINK. We also generated a separate file linking the lead or representative COG presence/absence pattern to tagged COGs with similar patterns.

**Generating the presence/absence patterns for unitigs**. Unique k-mers were identified in each draft genome assembly using DSK v2.3.0 (*-abundance-min-threshold 1 -abundance-min 1 -kmer-size 31 -solid-kmers-out*)[114]. The unitigs are computationally efficient and most importantly identify variants across the entire genome without the requirement of using a reference genome. We then used Bifrost[115] to construct a compact De Bruijn graph and generate unique maximal length unitigs; sequences represented by non-branching paths in the graph. The 909 genomes were decomposed into 4,122,999 k-mers of length 31 bp using DSK[114]. The mean number of unique k-mers per genome was 2,027,332 (range: 1,804,558 to 2,119,269). We converted the k-mers into a De Bruijn graph and then identified 123,401 unitigs[115]. The unitigs in Fasta format were compared to the a De Bruijn graph of each genome to generate a unitig presence/absence matrix[115]. The unitigs with MAF < 1% were excluded from the matrix and the resultant matrix was converted to the pedigree formatted files containing phenotype data similar to those required by PLINK.

**GWAS analysis of SNPs, accessory COGs and unitigs**. We used the FaST-LMM v2.07.20140723[55] and GEMMA v0.98[56] to fit a univariate linear mixed model for the association between SNPs, accessory COGs and unitigs with the disease phenotype of the isolates namely CNS and non-CNS. The input pedigree files were formatted as haploid human mitochondrial genotypes with chromosome code 'MT' similar to bacterial GWAS elsewhere[35,44]. We calculated the genetic relatedness matrix for FaST-LMM and GEMMA using the SNPs to control for population structure. The GWAS analysis included host age (years) and country of isolate origin covariates. We used Bonferroni correction to control for the false discovery rate due to multiple testing ($\alpha/N$) where the statistical significance level $\alpha$ and $N$ was the total number of SNPs, COGs and unitigs. The value of $\alpha$ was 0.05 and the value of $N$ for SNPs, unitigs and COGs were 2393, 20,673 and 1068 resulting in the genome-wide significant threshold ($0.05/N$) of $2.09 \times 10^{-05}$, $2.42 \times 10^{-06}$ and $4.68 \times 10^{-05}$ and suggestive threshold ($1/N$) of $4.18 \times 10^{-04}$, $4.84 \times 10^{-05}$ and $9.36 \times 10^{-04}$, respectively. The proportion of phenotypic variation explained by the genetics or the narrow-sense heritability was estimated using GEMMA. The genome-wide and suggestive unitigs were annotated using nucleotide-BLAST v2.2.30 + (match identity: 90%, query coverage: 90%)[108] and complete genome references of *S. pneumoniae* obtained from GenBank (Supplementary Table 1). The likelihood ratio test *P*-values from both FaST-LMM and GEMMA were used to generate the Manhattan plots and QQ-plots using 'ggplot2' v3.1.0[116]. Correlation between variants was assessed using 'ggcorrplot' v0.1.3 in R. Protein structures were modelled using SWISS-MODEL[117] and Robetta[118], and visualised using PyMOL v2.4.0[119].

**Reporting summary**. Further information on research design is available in the Nature Research Reporting Summary linked to this article.

## Data availability

The sequence data used in this study was deposited in the European Nucleotide Archive (ENA) and the accession numbers, isolate information and other source data underlying plots shown in main text figures are provided in Supplementary Data 1–3. The authors declare that all other data supporting the findings of this study are available within the paper and its Supplementary information files.

## Code availability

All tools and R packages used for the analysis are publicly available and fully described in the 'Methods' sections.

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

## Acknowledgements

We thank the clinical and laboratory teams at the collaborating institutions, and the sequencing and informatics teams at the Wellcome Sanger Institute. We are also grateful for the insightful feedback on the manuscript provided by Dr. Bernard Beall and Dr. Allen S. Craig at the Centers for Disease Control and Prevention (CDC) in the US. This study was funded by the Bill and Melinda Gates Foundation (grant number: OPP1023440 and OPP1034556). C.C., G.T. and S.D.B. were supported by funding from the Joint Programme Initiative for Antimicrobial Resistance (JPIAMR). The contents of this paper are solely the responsibility of the authors and does not necessarily represent the official views of their affiliated institutions and the funding agencies.

## Author contributions

Conceptualisation and study design: C.C. and S.D.B. Sample collection, microbiology and molecular work: D.B.E., J.E.C., C.P., C.E., G.P., S.O., R.S.H., N.F., A.V.G., M.D.P., M.A., B.A. K.A., B.S. and J.M.C. Whole-genome sequencing: SDB. Data curation and quality checks: C. C., J.E.C., S.W.L. and R.A.G. Bioinformatics and statistical analysis: C.C. Suggestions on GWAS analysis: G.T. Drafting initial manuscript: C.C. and S.D.B. Discussion and interpretation of the findings, and review and editing of the manuscript: C.C., M.Y., J.E.C., S.W. L., R.A.G., J.E.C., B.S., M.A., C.P., G.T., G.P., M.D.P., M.S., S.O., B.A.K.A., S.O., J.M.C., A.V. G., N.F., R.S.H., R.F.B., L.M., R.F.B., K.P.K., D.B.E., A.K. and S.D.B.

## Competing interests

The authors declare no competing interests.

## Additional information

Chrispin Chaguza [1,2✉], Marie Yang[3], Jennifer E. Cornick[3,4], Mignon du Plessis [5,6], Rebecca A. Gladstone[1], Brenda A. Kwambana-Adams[7,8], Stephanie W. Lo[1], Chinelo Ebruke[8], Gerry Tonkin-Hill [1], Chikondi Peno[4,9], Madikay Senghore[8,10], Stephen K. Obaro[11,12], Sani Ousmane[13], Gerd Pluschke[14], Jean-Marc Collard[13], Betuel Sigaùque[15], Neil French[3], Keith P. Klugman[16], Robert S. Heyderman[4,7], Lesley McGee [17], Martin Antonio[8,18], Robert F. Breiman[19], Anne von Gottberg[5,6], Dean B. Everett[4,9], Aras Kadioglu[3] & Stephen D. Bentley [1,20✉]

[1]Parasites and Microbes Programme, Wellcome Sanger Institute, Wellcome Genome Campus, Cambridge, UK. [2]Darwin College, University of Cambridge, Silver Street, Cambridge, UK. [3]Department of Clinical Infection, Microbiology and Immunology, Institute of Infection, Veterinary and Ecological Sciences, University of Liverpool, Liverpool, UK. [4]Malawi-Liverpool-Wellcome Trust Clinical Research Programme, Blantyre, Malawi. [5]Centre for Respiratory Diseases and Meningitis, National Institute for Communicable Diseases, Johannesburg, South Africa. [6]School of Pathology, Faculty of Health Sciences, University of the Witwatersrand, Johannesburg, South Africa. [7]NIHR Global Health Research Unit on Mucosal Pathogens, Division of Infection and Immunity, University College London, London, UK. [8]Medical Research Council (MRC) Unit The Gambia at the London School of Hygiene and Tropical Medicine, Fajara, The Gambia. [9]MRC Centre for Inflammation Research, Queens Medical Research Institute, University of Edinburgh, Edinburgh, UK. [10]Center for Communicable Disease Dynamics, Department of Epidemiology, Harvard TH Chan School of Public Health, Boston, MA, USA. [11]Division of Pediatric Infectious Disease, University of Nebraska Medical Center Omaha, Omaha, NE, USA. [12]International Foundation against Infectious Diseases in Nigeria, Abuja, Nigeria. [13]Centre de Recherche Médicale et Sanitaire, Niamey, Niger. [14]Swiss Tropical and Public Health Institute, Basel, Switzerland. [15]Centro de Investigação em Saúde da Manhiça, Maputo, Mozambique. [16]Hubert Department of Global Health, Rollins School of Public Health, Emory University, Atlanta, GA, USA. [17]Respiratory Diseases Branch, Centers for Disease Control and Prevention, Atlanta, GA, USA. [18]Warwick Medical School, University of Warwick, Coventry, UK. [19]Emory Global Health Institute, Emory University, Atlanta, GA, USA. [20]Department of Pathology, University of Cambridge, Cambridge, UK. ✉email: cc19@sanger.ac.uk; sdb@sanger.ac.uk

