## [Peer Review File · Communications Biology]

Reviewers' comments:

Reviewer #1 (Remarks to the Author):

The hypothesis tested in this study was that allelic variation in selected genes explain the neurotropism of some clones of *Streptococcus pneumoniae*. This was tested by detailed comparative genomic analysis of 909 invasive isolates of the hyper-virulent pneumococcal serotype 1, all from the African continent. The strains were collected from patients with infections in the central nervous system (CNS) i.e. cerebrospinal fluid (CSF), (N =297) and in non-CNS tissues (blood, lungs, joints, peritoneum)(N=612) from hospitalized patients of any age through hospital-based surveillance in West Africa. To avoid bias isolates from known outbreaks were not included. Extracted DNA from the strains was sequenced at the Wellcome Sanger Institute, UK, using Genome Analyser II and HiSeq 4000 Sequencing Systems. Appropriate quality control criteria were implemented. Based on sequence analyses each strain was assigned to serotype, MLST sequence type (ST), and international pneumococcal sequence cluster (GPSC) and the entire collection was subjected to phylogenetic analysis. State-of-the-art genetic and statistical analyses identified two genome-wide significant unitigs whose presence/absence were associated with CNS isolates: the genes encoding pneumococcal surface protein C (pspC)/ choline-binding protein A (cpbA), and the putative DnaQ associated with the putative DnaQ family exonuclease or DinG family helicase, showing odds ratios 0.70 and 0.71, respectively. In addition, above the threshold (odds ratio: 1.10) was the subunit S protein of a Type 1 restriction-modification system, previously shown to be an important regulator of capsule polysaccharide production, among other properties. Genome sequences are deposited in the European Nucleotide Archive (ENA).

The hypothesis is important and the study design is comprehensive and carefully performed. The results are interesting and of potential practical importance in attempts to prevent CNS infections. The manuscript is well written and the figures, including the supplementary figures, are of excellent standard.

A few points should be considered by the authors.

1. In addition to the properties listed, the pneumococcal surface protein C (pspC)/ choline-binding protein A (cpbA) was previously shown to bind the polymeric immunoglobulin receptor mediating trans-cellular transport (Hammerschmidt et al. *Mol Microbiol* 25:1113, 1997; Zhang et al. *Cell* 102:827, 2000) and is known to be up-regulated upon contact with epithelial cells (Orihuela et al. *Infect Immun* 72:5582, 2004). The corresponding gene is eliminated in *S. mitis* and other commensal streptococci (Kilian & Tettelin, *mBio* 2019; 10(5). pii: e01985-19. doi: 10.1128/mBio.01985-19), thus enhancing its potential use in a vaccine.
2. The mechanism of action of the Type 1 restriction-modification system and the demonstrated regulation of capsule production and other important properties (Manso et al. *Nat Commun* 5:5055. 2014.doi: 10.1038/ncomms6055) deserves to be mentioned.

Mogens Kilian

Reviewer #2 (Remarks to the Author):

This is an important, large scale study of a single serotype in the setting of endemic meningitis. The focus of the study on one serotype in a specific, large geographic region of study is novel and allows comparisons to be made to studies done in other geographical regions. The study aims to determine genetic variants associated with neurotropism that could be used for therapeutic or prophylactic interventions. The study succeeds in its aims to determine variants that may be associated with CNS infection but concedes that the biological picture is more complicated than a simple variant, as the effect size of the variants discovered is small and they do not show heritability. The results are still important, however, as the variants discovered are significant and can be the subject of further work.

I am by no-means an expert in the statistical and bioinformatic methods used, however it appears to me that they are appropriate and have been carefully controlled to reduce confounding influences. A significant amount of detail and supporting data is included which should allow other researchers to replicate the work. Limitations of the study are clearly discussed, for example the possibility of non-CNS isolates simply being "in transit" to the CNS.

I only have a few minor points to raise.

Figure 1a, while a nice way of presenting the study design, I'm unsure whether it shows anything that is not already described in the text. To me it is unclear whether the sizes or positions of the squares are meant to represent anything??

Line 209 references Figure 3A. the paragraph is discussing significant unitig associated with CNS isolates but the figure referenced is the number of snps vs clade, the figure reference is meant to be 4A.

Figure 4A. Perhaps does not need the x-axis scale as at first I thought, because it was labelled unitig (ID), that these were the ID numbers. But actually they are just a count of the unordered unitigs and the (ID) was referring to the annotated ID numbers.

Table 1 header - should it be unitigs rather than SNPS?

Line 359 - should refer to Table 2.

Fig S7 - The key is not shown, however I assume the key is the same as figure S6(b)

Reviewer #3 (Remarks to the Author):

General Comments:

After a careful and critical reading of this research study, it was easy to understand that the authors (Chaguza, et al.) are reporting for the first time important genetic associations modulating the tropism of the hyper-virulent *Streptococcus pneumoniae* serotype 1 into cerebrospinal fluid to cause central nervous system (CNS) infections, particularly meningitis.

The authors compared 909 pneumococcal serotype 1 strains isolated from CNS (297) and non-CNS (612) human specimens, aiming to determine whether the presence of genetic variation(s) may contribute to our understanding of the ability of the pneumococcus to translocate across the blood-brain-barrier to cause meningitis (neurotropism).

The manuscript provides solid GWAS data and analysis of these pneumococcal serotype 1 isolates collected in sub-Saharan countries during 20 years (1996-2016). However, the impact of pneumococcal conjugate vaccine (PCV) introduction (selective pressure) is not considered in the analysis. Serotype 1 is included in the PCV10 and PCV13 but was not included in the precedent PCV7 formulation. Is there any metadata about the vaccination status (post-PCV period) of the participants from where pneumococci were recovered?

Moreover, the most relevant result links a genetic variant (Unitig 8805) located in the PspC (CbpA, SpsA) coding region of the pneumococcal serotype 19A (Hungary 19A-14) genome with neurotropism, which is particularly in good agreement with the previous work of the co-author Dr. Aras Kadioglu(1) and other scientists in the field worldwide(2-6). A (supplementary) figure depicting the capsule 1 operon and the most relevant unitigs on a serotype 1 representative genome may be helpful. To highlight the pspC gene, PspC protein, Proline-rich domain and the

unitig 8805 would contribute to a better understanding of the discussion at a molecular level. Especially on how human antibodies would have protective effects and how its presence/absence can be beneficial/detrimental for a serotype 1 colonized person. Furthermore, PspC is a well distributed but highly variable gene and protein(7). Could the authors please show how conserved/variable the unitig 8805 (among their 909 sequenced serotype 1 strains) is?. This is especially important in order to conclude or suggest the usefulness of this protein region for the design of protein-based vaccines to prevent pneumococcal CNS infections(8). Finally, the authors describe clearly laboratory methods, platforms, pipelines and statistical analysis that will be useful to the scientific community in terms of comparability and reproducibility.

Minor Comments:

1. Introduction: When possible, it should consider short info about PCV vaccination in south-Saharan countries.
2. Page 7, Line 213: "...protective antibody-mediated protection"?
3. Results: This research study provides important findings to be shared with the scientific community.
4. Discussion: Should consider vaccination status, distribution and conservation of unitig 8805.
5. Figures: Please check x and y coordinates in Figure 5C and 5C (Page 26), as of Figure S7.
6. A serotype 1 representative genome and PspC protein structure should be include it for better understanding of the molecular findings. Please, show were the relevant unitigs are located.

References:

1. Jacques, L. C. et al. Increased pathogenicity of pneumococcal serotype 1 is driven by rapid autolysis and release of pneumolysin. *Nat. Commun.* 11, 1892 (2020).
2. Mook-Kanamori, B. B., Geldhoff, M., Poll, T. van der & Beek, D. van de. Pathogenesis and Pathophysiology of Pneumococcal Meningitis. *Clin. Microbiol. Rev.* 24, 557 (2011).
3. Orihuela, C. et al. Laminin receptor initiates bacterial contact with the blood brain barrier in experimental meningitis models. *The Journal of clinical investigation* vol. 119 <https://pubmed.ncbi.nlm.nih.gov/19436113/> (2009).
4. Dave, S., Carmicle, S., Hammerschmidt, S., Pangburn, M. K. & McDaniel, L. S. Dual Roles of PspC, a Surface Protein of *Streptococcus pneumoniae*, in Binding Human Secretory IgA and Factor H. *J. Immunol.* 173, 471–477 (2004).
5. Hammerschmidt, S., Talay, S. R., Brandtzaeg, P. & Chhatwal, G. S. SpsA, a novel pneumococcal surface protein with specific binding to secretory immunoglobulin A and secretory component. *Mol. Microbiol.* 25, 1113–1124 (1997).
6. Daniels, C. et al. The proline-rich region of pneumococcal surface proteins A and C contains surface-accessible epitopes common to all pneumococci and elicits antibody-mediated protection against sepsis. *Infection and immunity* vol. 78 <https://pubmed.ncbi.nlm.nih.gov/20194601/> (2010).
7. Gámez, G. et al. The variome of pneumococcal virulence factors and regulators. *BMC Genomics* 19, 1–18 (2018).
8. Gámez, G. & Hammerschmidt, S. Combat pneumococcal infections: adhesins as candidates for protein-based vaccine development. *Curr. Drug Targets* 13, 323–337 (2012).

Reviewers' comments:

Reviewer #1 (Remarks to the Author):

The hypothesis tested in this study was that allelic variation in selected genes explain the neurotropism of some clones of *Streptococcus pneumoniae*. This was tested by detailed comparative genomic analysis of 909 invasive isolated of the hyper-virulent pneumococcal serotype 1, all from the African continent. The strains were collected from patients with infections in the central nervous system (CNS) i.e. cerebrospinal fluid (CSF), (N =297) and in non-CNS tissues (blood, lungs, joints, peritoneum) (N=612) from hospitalized patients of any age through hospital-based surveillance in West Africa. To avoid bias isolates from known outbreaks were not included. Extracted DNA from the strains was sequenced at the Wellcome Sanger Institute, UK, using Genome Analyser II and HiSeq 4000 Sequencing Systems. Appropriate quality control criteria were implemented. Based on sequence analyses each strain was assigned to serotype, MLST sequence type (ST), and international pneumococcal sequence cluster (GPSC) and the entire collection was subjected to phylogenetic analysis. State-of-the-art genetic and statistical analyses identified two genome-wide significant unitigs whose presence/absence were associated with CNS isolates: the genes encoding pneumococcal surface protein C (pspC)/ choline-binding protein A (cpbA), and the putative DnaQ associated with the putative DnaQ family exonuclease or DinG family helicase, showing odds ratios 0.70 and 0.71, respectively. In addition, above the threshold (odds ratio: 1.10) was the subunit S protein of a Type 1 restriction-modification system, previously shown to be an important regulator of capsule polysaccharide production, among other properties. Genome sequences are deposited in the European Nucleotide Archive (ENA). The hypothesis is important, and the study design is comprehensive and carefully performed. The results are interesting and of potential practical importance in attempts to prevent CNS infections. The manuscript is well written and the figures, including the supplementary figures, are of excellent standard.

Response: Thank you for the excellent summary of the findings and their relevance as well as limitations of the paper.

A few points should be considered by the authors.

1. In addition to the properties listed, the pneumococcal surface protein C (pspC)/ choline-binding protein A (cpbA) was previously shown to bind the polymeric immunoglobulin receptor mediating trans-cellular transport (Hammerschmidt et al. Mol Microbiol 25:1113, 1997; Zhang et al. Cell 102:827, 2000) and is known to be up-regulated upon contact with epithelial cells (Orihuela et al. Infect Immun 72:5582, 2004). The corresponding gene is eliminated in *S. mitis* and other commensal streptococci (Kilian & Tettelin, mBio 2019; 10(5). pii: e01985-19. doi: 10.1128/mBio.01985-19), thus enhancing its potential use in a vaccine.

Response: Thank you for your excellent suggestions on the literature regarding the role of PspC. We have now incorporated this information and cited the suggested references in the manuscript in lines 303-307 and 313-315.

2. The mechanism of action of the Type 1 restriction-modification system and the

demonstrated regulation of capsule production and other important properties (Manso et al. Nat Commun 5:5055. 2014.doi: 10.1038/ncomms6055) deserves to be mentioned.

Response: We have now mentioned the suggested mechanism of action for the Type 1 restriction modification system in the revised manuscript in line 241.

Mogens Kilian

Reviewer #2 (Remarks to the Author):

This is an important, large scale study of a single serotype in the setting of endemic meningitis. The focus of the study on one serotype in a specific, large geographic region of study is novel and allows comparisons to be made to studies done in other geographical regions. The study aims to determine genetic variants associated with neurotropism that could be used for therapeutic or prophylactic interventions. The study succeeds in its aims to determine variants that may be associated with CNS infection but concedes that the biological picture is more complicated than a simple variant, as the effect size of the variants discovered is small and they do not show heritability. The results are still important, however, as the variants discovered are significant and can be the subject of further work.

I am by no means an expert in the statistical and bioinformatic methods used, however it appears to me that they are appropriate and have been carefully controlled to reduce confounding influences. A significant amount of detail and supporting data is included which should allow other researchers to replicate the work. Limitations of the study are clearly discussed, for example the possibility of non-CNS isolates simply being "in transit" to the CNS.

Response: Thank you for the excellent summary of the findings and their relevance as well as limitations of the paper.

I only have a few minor points to raise.

1. Figure 1a, while a nice way of presenting the study design, I'm unsure whether it shows anything that is not already described in the text. To me it is unclear whether the sizes or positions of the squares are meant to represent anything?

Response: This is an excellent assessment of the figure. The squares in Fig. 1a are of the same size but placed at different positions as each line represents different data types used in the GWAS analysis namely SNPs, unitigs and gene presence/absence. We have now labelled the squares to show the data types that they represent for clarity.

2. Line 209 references Figure 3A. the paragraph is discussing significant unitig associated with CNS isolates, but the figure referenced is the number of snps vs clade, the figure reference is meant to be 4A.

Response: Indeed, the reference was meant for Fig. 4a instead of Fig. 3a. We have now amended the text to reference Fig. 4a.

3. Figure 4A. Perhaps does not need the x-axis scale as at first I thought, because it was labelled unitig (ID), that these were the ID numbers. But actually, they are just a count of the unordered unitigs and the (ID) was referring to the annotated ID numbers.

Response: We concur with the reviewer's assessment. We have now removed the x-axis scale, which in this case was unnecessary, to avoid misunderstanding as pointed out by the reviewer.

4. Table 1 header - should it be unitigs rather than SNPs?

Response: Yes, Table 1's header should be unitigs rather than SNPs. We have now corrected text.

5. Line 359 - should refer to Table 2.

Response: We have now added a reference to Table 2 at the suggested location corresponding to line 372 in the revised manuscript.

6. Fig S7 - The key is not shown; however, I assume the key is the same as figure S6(b)

Response: Thank you for highlighting this. We have now added the key to Supplementary Fig. 8 (previously Fig. S7).

Reviewer #3 (Remarks to the Author):

General Comments: After a careful and critical reading of this research study, it was easy to understand that the authors (Chaguza, et al.) are reporting for the first time important genetic associations modulating the tropism of the hyper-virulent *Streptococcus pneumoniae* serotype 1 into cerebrospinal fluid to cause central nervous system (CNS) infections, particularly meningitis. The authors compared 909 pneumococcal serotype 1 strains isolated from CNS (297) and non-CNS (612) human specimens, aiming to determine whether the presence of genetic variation(s) may contribute to our understanding of the ability of the pneumococcus to translocate across the blood-brain-barrier to cause meningitis (neurotropism). The manuscript provides solid GWAS data and analysis of these pneumococcal serotype 1 isolates collected in sub-Saharan countries during 20 years (1996-2016). However, the impact of pneumococcal conjugate vaccine (PCV) introduction (selective pressure) is not considered in the analysis. Serotype 1 is included in the PCV10 and PCV13 but was not included in the precedent PCV7 formulation. Is there any metadata about the vaccination status (post-PCV period) of the participants from where pneumococci were recovered? Moreover, the most relevant result links a genetic variant (Unitig 8805) located in the PspC (CbpA, SpsA) coding region of the pneumococcal serotype 19A (Hungary 19A-14) genome with neurotropism, which is particularly in good agreement with the previous work of the co-author Dr. Aras Kadioglu (1) and other scientists in the field

worldwide (2–6). A (supplementary) figure depicting the capsule 1 operon and the most relevant unitigs on a serotype 1 representative genome may be helpful. To highlight the *pspC* gene, PspC protein, Proline-rich domain and the unitig 8805 would contribute to a better understanding of the discussion at a molecular level. Especially on how human antibodies would have protective effects and how its presence/absence can be beneficial/detrimental for a serotype 1 colonized person. Furthermore, PspC is a well distributed but highly variable gene and protein (7). Could the authors please show how conserved/variable the unitig 8805 (among their 909 sequenced serotype 1 strains) is?. This is especially important in order to conclude or suggest the usefulness of this protein region for the design of protein-based vaccines to prevent pneumococcal CNS infections (8). Finally, the authors describe clearly laboratory methods, platforms, pipelines and statistical analysis that will be useful to the scientific community in terms of comparability and reproducibility.

Response: Thank you for your excellent summary of the paper and comments. The majority of the isolates were collected prior to the introduction of 10- or 13-valent pneumococcal conjugate vaccines (PCV) in Sub Saharan Africa. Since sampling of the isolates was across all age groups, the majority of the isolates used in the analysis were from unvaccinated individuals. It is possible that some of the isolates may have come from vaccinated infants, but this would have been a small subset of the dataset for the reasons mentioned although the individual vaccination status of infants sampled after vaccination is not available. As such, we are of the view that the pneumococcal vaccination is highly unlikely to have biased our GWAS findings considering that the PCVs targets the polysaccharide capsule, which was identical in all the isolates analysed, rather than certain protein variants outside the capsule such as those identified by our analysis. We have now highlighted this information in lines 376-381 and provided year of isolation for the isolates in Supplementary Data 1. As suggested, we have now included data showing the conservation of the genomic region containing the unitig 8805 in line 313-316 and Supplementary Data 3 while the distribution of the unitig 8805 across the phylogenetic tree is shown in Fig. 5b. Furthermore, we have a diagram showing locations of the capsule 1 operon/locus and the genome-wide significant unitigs has been included in Fig. 5a as suggested.

Minor Comments:

1. Introduction: When possible, it should consider short info about PCV vaccination in south-Saharan countries.

Response: We concur with the reviewer’s suggestion. We have now added some information on PCV vaccination in Sub Saharan African countries in the introduction section in lines 79-81 and discussion section in lines 287-289 and 413-415.

2. Page 7, Line 213: “...protective antibody-mediated protection”?

Response: Thank you for pointing out this error. We have now revised the sentence from “...protective antibody-mediated protection” to “...antibody-mediated protection”.

3. Results: This research study provides important findings to be shared with the scientific community.

Response: Thank you for your assessment of the study findings and their relevance.

4. Discussion: Should consider vaccination status, distribution and conservation of unitig 8805.

Response: We have now included a brief discussion of vaccination status of the isolates and rollout pneumococcal conjugate vaccines in Sub Saharan Africa (lines 80-82, 287-289 and 413-415) in addition to statements on the phylogenetic and geographical distribution, and conservation of the unitig 8805 across the isolates. As suggested, we have analysed the sequence conservation of the genomic region containing the unitig 8805 to understand its conservation in the isolates (Supplementary Data 3 and line 313-316).

5. Figures: Please check x and y coordinates in Figure 5C and 5C (Page 26), as of Figure S7.

Response: The x- and y-labels were switched in the figure. We have now amended all the figures to show correct labels.

6. A serotype 1 representative genome and PspC protein structure should be included for better understanding of the molecular findings. Please, show where the relevant unitigs are located.

Response: This is an interesting suggestion. As suggested, we have now included a serotype 1 representative genome (Fig. 5a) and indicated where the relevant (genome-wide significant) unitigs are located. Furthermore, we have included predicted protein structure for *pspC* and the putative exonuclease, which harboured the genome-wide significant unitigs, in Supplementary Fig. 5.

References:

1. Jacques, L. C. et al. Increased pathogenicity of pneumococcal serotype 1 is driven by rapid autolysis and release of pneumolysin. *Nat. Commun.* 11, 1892 (2020).
2. Mook-Kanamori, B. B., Geldhoff, M., Poll, T. van der & Beek, D. van de. Pathogenesis and Pathophysiology of Pneumococcal Meningitis. *Clin. Microbiol. Rev.* 24, 557 (2011).
3. Orihuela, C. et al. Laminin receptor initiates bacterial contact with the blood brain barrier in experimental meningitis models. *The Journal of clinical investigation* vol. 119 <https://pubmed.ncbi.nlm.nih.gov/19436113/> [pubmed.ncbi.nlm.nih.gov] (2009).
4. Dave, S., Carmicle, S., Hammerschmidt, S., Pangburn, M. K. & McDaniel, L. S. Dual Roles of PspC, a Surface Protein of *Streptococcus pneumoniae*, in Binding Human Secretory IgA and Factor H. *J. Immunol.* 173, 471–477 (2004).
5. Hammerschmidt, S., Talay, S. R., Brandtzaeg, P. & Chhatwal, G. S. SpsA, a novel pneumococcal surface protein with specific binding to secretory immunoglobulin A and secretory component. *Mol. Microbiol.* 25, 1113–1124 (1997).

6. Daniels, C. et al. The proline-rich region of pneumococcal surface proteins A and C contains surface-accessible epitopes common to all pneumococci and elicits antibody-mediated protection against sepsis. *Infection and immunity* vol. 78 <https://pubmed.ncbi.nlm.nih.gov/20194601/> [pubmed.ncbi.nlm.nih.gov] (2010).
7. Gámez, G. et al. The variome of pneumococcal virulence factors and regulators. *BMC Genomics* 19, 1–18 (2018).
8. Gámez, G. & Hammerschmidt, S. Combat pneumococcal infections: adhesins as candidates for protein-based vaccine development. *Curr. Drug Targets* 13, 323–337 (2012).

REVIEWERS' COMMENTS:

Reviewer #3 (Remarks to the Author):

After a careful and critical inspection of the answers given by the authors to my suggestions and comments, I could see they were appropriately addressed.

The additional info provided for the main point, concerning the impact of the PCV conjugate vaccine on the genomic landscape of the pneumococcal serotype 1 genome, is in good agreement with previous research reports (Mackenzie, 2016 and Cohen, 2017), as well as with the previous work of one of the Co-Authors: Ebruke, CN; (2018) Molecular epidemiological and pathogenesis studies of *Streptococcus pneumoniae* serotype 1 strains from West Africa. MPhil thesis, London School of Hygiene & Tropical Medicine. DOI: (<https://doi.org/10.17037/PUBS.04647232>). The discussion was appropriately addressed and it enhances the manuscript, highlighting the need for alternative (capsule-independent / protein-based) strategies to fight against this human pathogen.

The second point, about the distribution and conservation status of the genomic region containing the unitig 8805, is also appropriately addressed, providing good data and relevant discussion.

However, a couple of molecular refinements are further suggested in order to improve the final version of this nice piece of work:

1. Regarding the serotype 1 reference genome sequence (Spn1041): The NCBI entry CACE01000000 is "the master record for a whole genome shotgun sequencing project and contains no sequence data". (<https://www.ncbi.nlm.nih.gov/nucleotide/CACE01000000>). Difficult to follow... ..better P1031 or gamPNI0373???
2. I am not sure about the chromosomal location of *pspC* in your Fig. 5a. Please, check some examples I respectfully prepared for your guidance (Slide-1). Furthermore, it is not necessary to display the serotype 1 capsule operon/locus from another serotype 1 genome (*Streptococcus pneumoniae* strain 519/43 - CR931632.1).
3. Please, check the protein nomenclature in the Supplementary Fig. 5. *PspC* and *CbpA* are for proteins, whereas *pspC* and *cbpA* are for genes. Moreover, it would be really elegant to highlight the in frame-translated protein sequences of your unitigs in your predicted protein structures (Slide-2).

Reviewers' comments:

Reviewer #3 (Remarks to the Author):

After a careful and critical inspection of the answers given by the authors to my suggestions and comments, I could see they were appropriately addressed.

The additional info provided for the main point, concerning the impact of the PCV conjugate vaccine on the genomic landscape of the pneumococcal serotype 1 genome, is in good agreement with previous research reports (Mackenzie, 2016 and Cohen, 2017), as well as with the previous work of one of the Co-Authors: Ebruke, CN; (2018) Molecular epidemiological and pathogenesis studies of *Streptococcus pneumoniae* serotype 1 strains from West Africa. MPhil thesis, London School of Hygiene & Tropical Medicine. DOI: (<https://doi.org/10.17037/PUBS.04647232> [doi.org]). The discussion was appropriately addressed and it enhances the manuscript, highlighting the need for alternative (capsule-independent / protein-based) strategies to fight against this human pathogen.

The second point, about the distribution and conservation status of the genomic region containing the unitig 8805, is also appropriately addressed, providing good data and relevant discussion.

However, a couple of molecular refinements are further suggested in order to improve the final version of this nice piece of work:

1. Regarding the serotype 1 reference genome sequence (Spn1041): The NCBI entry CACE01000000 is "the master record for a whole genome shotgun sequencing project and contains no sequence data". (<https://www.ncbi.nlm.nih.gov/nuccore/CACE01000000> [ncbi.nlm.nih.gov]). Difficult to follow... ..better P1031 or gamPNI0373???

Response: Thank you for this suggestion which is important for reproducibility of the findings. We have updated the SNP-based GWAS analysis, which previously used the SNP calls after mapping to the Spn1041 reference genome. As suggested, we have now used the pneumococcal serotype 1 reference genome for strain P1031 (GenBank accession: CP000920) and updated the results and figures accordingly.

2. I am not sure about the chromosomal location of *pspC* in your Fig. 5a. Please, check some examples I respectfully prepared for your guidance (Slide-1). Furthermore, it is not necessary to display the serotype 1 capsule operon/locus from another serotype 1 genome (*Streptococcus pneumoniae* strain 519/43 - CR931632.1).

Response: Thank you for the suggestion. We have now updated Fig. 5a to show the correct location of *pspC*. We have also shown the capsule operon in reference genome for strain P1031 (GenBank accession: CP000920). For clarity, we have labelled the capsule operon genes in strain P1031 using the common gene names similar to those used in *Streptococcus pneumoniae* strain 519/43 (GenBank accession: CR931632.1).

3. Please, check the protein nomenclature in the Supplementary Fig. 5. PspC and CbpA are for

proteins, whereas *pspC* and *cbpA* are for genes. Moreover, it would be really elegant to highlight the in frame-translated protein sequences of your unitigs in your predicted protein structures (Slide-2).

Response: Thank you for pointing this out. We have corrected the protein nomenclature and highlighted the location of the unitig sequences in the predicted protein structures in Supplementary Fig. 5 as suggested.